# The Neurokinin-1 Receptor Antagonist Aprepitant, a New Drug for the Treatment of Hematological Malignancies: Focus on Acute Myeloid Leukemia

**DOI:** 10.3390/jcm9061659

**Published:** 2020-06-01

**Authors:** Miguel Muñoz, Rafael Coveñas

**Affiliations:** 1Research Laboratory on Neuropeptides (IBIS), Virgen del Rocío University Hospital, 41013 Sevilla, Spain; 2Institute of Neurosciences of Castilla y León (INCYL), Laboratory of Neuroanatomy of the Peptidergic Systems, University of Salamanca, 37007 Salamanca, Spain; covenas@usal.es

**Keywords:** substance P, NK-1R, AML, aprepitant, anti-leukemic, apoptosis, non-solid tumor

## Abstract

Acute myeloid leukemia (AML) is a heterogeneous hematological malignancy. To treat the disease successfully, new therapeutic strategies are urgently needed. One of these strategies can be the use of neurokinin-1 receptor (NK-1R) antagonists (e.g., aprepitant), because the substance P (SP)/NK-1R system is involved in cancer progression, including AML. AML patients show an up-regulation of the NK-1R mRNA expression; human AML cell lines show immunoreactivity for both SP and the NK-1R (it is overexpressed: the truncated isoform is more expressed than the full-length form) and, via this receptor, SP and NK-1R antagonists (aprepitant, in a concentration-dependent manner) respectively exert a proliferative action or an antileukemic effect (apoptotic mechanisms are triggered by promoting oxidative stress via mitochondrial Ca^++^ overload). Aprepitant inhibits the formation of AML cell colonies and, in combination with chemotherapeutic drugs, is more effective in inducing cytotoxic effects and AML cell growth blockade. NK-1R antagonists also exert an antinociceptive effect in myeloid leukemia-induced bone pain. The antitumor effect of aprepitant is diminished when the NF-κB pathway is overactivated and the damage induced by aprepitant in cancer cells is higher than that exerted in non-cancer cells. Thus, the SP/NK-1R system is involved in AML, and aprepitant is a promising antitumor strategy against this hematological malignancy. In this review, the involvement of this system in solid and non-solid tumors (in particular in AML) is updated and the use of aprepitant as an anti-leukemic strategy for the treatment of AML is also mentioned (a dose of aprepitant (>20 mg/kg/day) for a period of time according to the response to treatment is suggested). Aprepitant is currently used in clinical practice as an anti-nausea medication.

## 1. Introduction

Human myeloid leukemia (acute or chronic) shows an abnormal expansion of white blood cells in both bone marrow and blood [1,2,3]. Acute myeloid leukemia (AML) is a heterogeneous disease and its treatment is difficult because the characteristics of the cytogenetic subtypes and the morphological and clinical properties can be very different from person to person [4]. AML shows two peaks in occurrence in early childhood and later adulthood. AML has a very poor outcome in adults, with the majority not surviving five years. The long-term disease-free survival of AML patients is poor, with minimal improvement over the past decades: the five-year survival rate for people (20 years and older) with AML is approximately 24%, whereas for those under 20, it is 67% [5]. The standard AML treatment uses anthracyclines combined with cytarabine, but despite post-remission therapies, less than half of AML patients are cured. Another therapeutic strategy is allogeneic hematopoietic stem cell transplantation which combines adoptive immunotherapy with cytoreductive chemotherapy. Although advances in AML treatment showed significant improvements in outcomes for younger patients, prognosis in the elderly, who account for the majority of new cases, remains poor [6]. Therefore, it is imperative to search for new therapeutic strategies to treat human myeloid leukemia and, in particular, AML.

The undecapeptide substance P (SP), originating from the *TAC1* gene, belongs to the tachykinin family of peptides. The undecapeptide can be processed, and biological active fragments (e.g., SP_1-4_, SP_1-7_) originate from it [7]. SP is widely distributed by the peripheral and central nervous systems and it has also been observed in dendritic cells, mast cells, monocytes, lymphocytes, eosinophils, macrophages, smooth muscle cells, fibroblasts and cancer cells as well as in body fluids (blood, cerebrospinal fluid, breast milk) [7,8,9]. Other members belonging to this family are hemokinin-1, neurokinin A/B, kassinin, ranakinin, eledoisin and neuropeptide K. These peptides (including SP) are involved, after binding to the metabotropic neurokinin (NK)-1, NK-2 and NK-3 receptors, in many physiological/pathophysiological processes (e.g., cancer, pruritus, emesis, inflammation, viral and bacterial infection, pain, alcohol addiction, depression, anxiety, hematopoiesis) [8,9,10]. These receptors belong to the G-protein-coupled receptor family and are encoded by *TACR1* (NK-1R)*, TACR2* (NK-2R) and *TACR3* (NK-3R) genes [9]. SP and hemokinin-1 are the natural ligands of the NK-1R, which contains seven hydrophobic alpha-helical transmembrane domains with three extracellular and three intracellular loops [9,11]. The activation of the NK-1R by SP induces a clathrin-dependent mechanism internalization of the NK-1R and the induction of cell signaling pathways (ROCK, protein kinases A/C and adenylyl cyclase are activated) promotes the synthesis of DNA, diacylglycerol, inositol triphosphate, transcription factors and pro-inflammatory cytokines and also exerts an anti-apoptotic action (Figure 1) [7,9,12]. Like SP, the NK-1R is widely distributed by the whole body: skin, lung, thymus, thyroid gland, genitourinary/gastrointestinal tracts, dendritic cells, leucocytes, macrophages, lymphocytes, endothelial cells, placenta, spleen, smooth muscle, peripheral and central nervous systems, salivary glands and lymph nodes [8,9]. 

Many studies have shown that the SP/NK-1R system is involved in cancer, that the NK-1R is a crucial target for the treatment of cancer (tumor cells overexpress the NK-1R) and that NK-1R antagonists are potential broad-spectrum antitumor drugs [for a review, see 13]. In fact, many data have shown that SP, via the NK-1R, promotes the proliferation, migration, invasion and metastasis of tumor cells; exerts an anti-apoptotic effect in these cells and favors angiogenesis to increase tumor development by increasing tumoral blood supply (Figure 1) [13,14,15,16,17]. In solid tumors, by suppressing the expression of vascular endothelial growth factor and hypoxia-inducible factor-1α, NK-1R antagonists block the SP-mediated growth of endothelial cells [18,19]. Consequently, the SP/NK-1R system is involved in the growth/development of tumors (not only in solid tumors, but in hematopoietic malignant cells as well) [13,20,21,22]. However, there are some data suggesting that SP, by stimulating the immune system, inhibits the growth of tumor cells [23,24]. SP regulates proto-oncogenes and transcription factors (hypoxia-inducible factor, c-myc, c-jun, AP-1, c-fos) involved in cell cycle progression, cellular transformation/differentiation and apoptosis [25,26]. 

Despite the abundance of available data on the involvement of the SP/NK-1R system in solid tumors, a relatively small amount of research has been performed on the involvement of this system in hematological malignancies. In adults, AML is the most common myeloid leukemia and new targets/therapeutic strategies are urgent needed for its successful treatment. As indicated above, one of these emerging systems is the SP/NK-1R system [13]. In this review, the main aim is to show the important role that SP and the NK-1R play in cancer, particularly in AML, and how NK-1R antagonists (e.g., aprepitant) could be used to treat hematological malignancies.

## 2. The SP/NK-1R System as a Predictive Factor in Cancer and the Use of the NK-1R as a Potential Tumor Biomarker

Currently, it is known that the SP/NK-1R system is up-regulated in cancer (Figure 1) [13]. As tumor cells express SP in the cytoplasm, the release of the undecapeptide from cancer cells could exert autocrine, paracrine and endocrine mechanisms on these cells (e.g., SP could be released into blood vessels, thereby increasing SP plasma level) [13,27]. The latter is important as a predictive factor because it could indicate tumor development and/or a high risk of developing cancer since the level of SP in serum (the undecapeptide induces the mitogenesis of tumor cells) and the number of NK-1Rs were increased in patients with cancer in comparison to healthy subjects [28,29]. Moreover, a higher activity of the protein kinase B has been related to poor prognosis, and it is known that SP increases the activity of this protein kinase (Figure 1) [30]. A correlation has also been reported between HER-2 and SP, but not between SP and lymph node involvement, estrogen/progesterone receptors and tumor grade/size [28]. This is important because the undecapeptide, via an autocrine signaling, is involved in an HER-2 persistent activation that favors malignant progression and drug resistance [31], meaning that the HER-2 steady state could be diminished by the inhibition of SP mediated effects [32].

Lymph node involvement and the cancer stage have been related to the expression of the NK-1R and, in fact, patients with poor prognosis and more advanced and less-differentiated tumors expressed a higher number of NK-1Rs [28,29,33]. This is an important observation, because it means that the NK-1R can be used in cancer as a biomarker and it could be useful for an earlier diagnosis/treatment of the disease [28]. Moreover, the NK-1R has been observed in the cytoplasm of tumor cells and it has been suggested that the cytoplasmic expression of this receptor is a crucial prognostic factor as, the NK-1R expression level has been related with tumor grade and tumor node metastasis [28]. The higher expression of the NK-1R has been related with poorer survival, larger tumor size and a higher invasion/metastatic potential [34,35].

## 3. The SP/NK-1R System and Cancer: Cell Signaling Pathways, Overexpression of the NK-1R, Metastasis and NK-1R Antagonists

In tumor cells, SP through the NK-1R promotes an anti-apoptotic effect as well as proliferation, migration, invasion and metastasis (Figure 1) [13,25,27,30,36,37]. Cancer cells express both SP and the NK-1R, and it seems that the undecapeptide is involved in an autocrine mechanism that promotes mitogenesis in tumor cells [13,21,28,38,39,40,41,42,43,44]. Cancer cell proliferation is activated via mitogen-activated protein kinases (MAPKs, including signal-regulated kinases 1 and 2 as well as p38MAPK) and then the expression of c-myc/c-fos is induced [25]; in these processes, the involvement of protein kinase C delta and Scr has been demonstrated (Figure 1) [45]. Moreover, SP induces the proliferation of tumor cells through the NK-1R/Hairy and Enhancer of Split 1 (Hes 1), a transcriptional inhibitor of the Notch signaling pathway involved in the appearance/development of tumors [34]. SP up-regulates Hes 1 expression, and it has been suggested that cancer cell proliferation is related to a downstream regulation of Hes 1; this transcriptional inhibitor reduced the growth suppression of tumor cells caused by a downregulation of the NK-1R (Figure 1) [34]. In cancer cells, it has recently been demonstrated that SP, through the NK-1R, induces the mammalian target of rapamycin (mTOR) signaling axis and increases cancer cell growth/metastasis by the activation of the eukaryotic initiation factor 4E-binding protein 1 and p70 S6 kinase (Figure 1) [20], whereas aprepitant (a NK-1R antagonist) attenuates the mTOR activation by decreasing the phosphorylation of p70 S6 kinase [46]. 

It is also known that tumor cells overexpress the NK-1R, and in vitro experiments have demonstrated that this receptor is essential for the viability of these cells [22,41,42,43,47]. Thus, many data have shown that the NK-1R is a new potential target for the treatment of any type of tumor (both solid and non-solid), because it is known that NK-1R antagonists, via the NK-1R, induce apoptosis in cancer cells (Figure 1) [1,13,21,27,30,37,48,49]. In this sense, recent therapeutic strategies in which the SP/NK-1R is involved have shown promising results. Thus, in a patient treated with radiotherapy and a NK-1R antagonist (aprepitant, 1140 mg/day for 45 days), the tumor mass disappeared after six months and no serious side-effects were observed [50], and in patients with cancer, the efficacy (promoting apoptosis) of targeted alpha therapy with ^213^Bi-DOTA-SP (radionuclide tumor therapy) against tumor cells has been reported [51,52,53,54,55]. Another important finding is that SP, in cancer cells, augmented the expression of the NK-1R, but not that of other receptors (e.g., NK-2R) [56]. This finding is crucial because SP, in addition to promoting mitogenesis/migration of cancer cells as well as an anti-apoptotic effect (that is exerting beneficial actions in tumor cells), increases the expression of the NK-1R, which is overexpressed in cancer cells. OK Overall, the data suggest that the NK-1R is an important target for cancer treatment (Figure 1).

It is also important to note that the SP/NK-1R system is involved in the migration of cancer cells (Figure 1) [36]. This mechanism is mediated by the Akt/NF-κB pathway: the undecapeptide increases tumor-associated cytokines and NF-κB p65 expression and decreases cancer cell proliferation/migration when a reduced activation of Akt/NF-κB occurs [36]. The SP/NK-1R system also controls membrane blebbing, a mechanism related to cell migration and spreading (Figure 1) [27,36,57,58]. Moreover, SP favors the expression of matrix metalloproteinases (degradative enzymes) by enhancing the activation of several signaling pathways (Akt, JNK, ERK1/2) and promoting cancer cell migration, invasion and metastasis [59,60], processes that are inhibited by NK-1R antagonists [36].

In in vitro and in vivo experiments, NK-1R antagonists blocked (in a concentration-dependent manner) the proliferation of tumor cells by inducing apoptotic mechanisms [13,18,27,37,50,61,62]; inhibiting the basal kinase activity/phosphorylation of Akt [30]; increasing the expression of p21 and p27 (cell cycle regulatory proteins) and inducing G1/S cell cycle arrest [63]. In summary, as any cancer cell overexpresses the NK-1R and SP promotes the mitogenesis of these cells, a common therapeutic strategy against any tumor type (solid and non-solid) is possible: the use of NK-1R antagonists (Figure 1). 

## 4. The NK-1R Is Essential for the Viability of Tumor Cells

In solid and non-solid tumors, the involvement of the NK-1R in the viability of cancer cells has been reported in in vitro experiments (e.g., acute lymphoblastic leukemia, breast cancer, lung cancer, melanoma) [22,41,42,43,47,49]. Thus, when the NK-1R expression was blocked by a knockdown gene silencing method, the number of cancer cells decreased (due to apoptotic mechanisms). In tumor cells, it has recently been demonstrated that the silencing of the NK-1R induced G2/M phase arrest/apoptosis and the inhibition of cancer cell proliferation; the same findings were observed when tumor cells were treated with aprepitant [49]. However, SP counteracted the effects (regarding cell proliferation/apoptosis) mediated by the NK-1R silencing procedure [49]. Thus, irrespective of the tumor type, a common anti-tumor strategy (pharmacological strategy (with NK-1R antagonists) or genetic strategy (knockdown gene silencing method)) could be applied to treat any tumor: in both cases, the same therapeutic effect can be reached (apoptosis of tumor cells) (Figure 1) [4,22,30,37,41,42,43,47]. Thus, both therapeutic strategies can improve prognosis/survival of patients with any type of cancer. It has been demonstrated that aprepitant induces apoptosis in cancer cells by increasing mitochondrial reactive oxygen species; thus, a Ca^2+^ flux from the endoplasmic reticulum into mitochondria occurs and hence an impairment of its function appears [1]. 

Why is the NK-1R essential for cancer cells? As tumor cells need the beneficial SP stimulus (which induces cell proliferation and migration, an anti-apoptotic effect and increases the synthesis of the NK-1R), they overexpress the NK-1R [13,18,27,56]. When the NK-1R expression is silenced, or NK-1R antagonists are administered (which leads to the blockade of the SP stimulus), tumor cells develop apoptotic mechanisms (Figure 1). Another beneficial effect of SP in tumor cells, via the NK-1R, is the induction of glycogen breakdown and then the glucose obtained by cancer cells is used to increase their metabolism [13,64]. The glycolytic rate of tumor cells is 200 times higher (known as the Warburg effect) than that observed in normal cells, and consequently, the NK-1R is also needed to obtain glucose and, in the case that the stimulus is blocked by NK-1R antagonists, tumor cells die by apoptotic mechanisms due to starvation and hence an anti-Warburg action occurs [13,64]. This is an important point because cancer cells may respond to drug treatment by up-regulating mechanisms (e.g., increasing glycolysis) to obtain ATP and thus cells unable to adapt to oxidative and energetic stresses will die. It has been also reported that blocking the SP stimulus with antibodies against SP led to an observed increase in apoptosis and decrease in cell survival within cancer cells [31,32]. In contrast to the NK-1R, SP is not essential for cancer cells, because the undecapeptide can be released from nerve cells (the number of SP-positive nerves is related to cancer differentiation) [65] and from immune cells (e.g., located in the tumor microenvironment), and SP can reach cancer cells from the blood and/or the NK-1R can be also activated by hemokinin-1, a peptide that belongs to the tachykinin family of peptides and that, in addition to SP, is a peripheral natural ligand of the NK-1R. SP and hemokinin-1 have the same C-terminal sequence (Gln-Phe-Phe-Gly-Leu-Met) and show a high similar affinity for the NK-1R. Both peptides, SP and hemokinin-1, promote the proliferation of tumor cells, stimulate Ca^++^ mobilization and increase cytokine mRNA expression (Figure 1) [66].

## 5. Acute Myeloid Leukemia: SP/NK-1R System and the NK-1R Antagonist Aprepitant

In bone marrow, AML is the lethal overgrowth of the myeloid progeny and a very challenging hematologic malignancy. As indicated above, the involvement of the SP/NK-1R system in cancer opens an antitumor promising research line by using NK-1R antagonists. In bone marrow, tachykinins (SP, neurokinin A) are released from nerve fibers and/or from stroma cells and after binding to their respective receptors modulate hematopoiesis: SP promotes hematopoietic stimulation by increasing bone marrow progenitor proliferation, and neurokinin A exerts an opposite effect (Table 1) [67,68,69]. In altered leukemia cells, SP (but not neurokinin A) is produced, and this has been related to bone marrow fibrosis and leukemia (Table 1) [70,71,72,73]. It is also important to note that human IM-9 B-lymphoblasts express a higher number of NK-1Rs per cell (25,000-30,000) than healthy T lymphocytes (7000-10,000) and that SP, via an autocrine mechanism, promotes the proliferation of basophilic leukemia cells [70,73,74]. In AML patients, an increased microvessel density has been reported in bone marrow, meaning that, in this disease, angiogenesis plays an important role and hence an anti-angiogenic strategy (using NK-1R antagonists) could be beneficial for the treatment of AML (Figure 1) [18,19,75]. Overall, the data show that the use of NK-1R antagonists is a promising antitumor strategy against hematological malignancies. 

Aprepitant (Emend, L-754,030, MK-869; oral administration); aprepitant injectable emulsion (Cinvanti); fosaprepitant dimeglumine (Ivemend; a water-soluble prodrug of aprepitant; intravenous administration) which is converted to aprepitant by ubiquitous phosphatases and rolapitant (Varubi) are non-peptide NK-1R antagonists used in clinical practice for the treatment of acute/delayed chemotherapy-induced nausea/vomiting and post-operative nausea/vomiting [76,77]. In adults, NK-1R antagonists are well tolerated and safe, even at high doses (300 mg/day; 1140 mg/day for 45 days), and a lack of serious side-effects has been confirmed [50,77,78]. However, one study has suggested an increase incidence of febrile neutropenia in pedriatic osteosarcoma [79]. In this study, by using the Naranjo score, adverse events were classified as probable or possible [79]. Further safety studies must be performed in this population to confirm such adverse events and to know the origin of this serious effect. In human liver, aprepitant is mainly metabolized by cytochrome P450, family 3, subfamily A (CYP3A4), it binds to plasma proteins and shows half-life ranges from 9 to 13 h [50,80,81]. It is important to comment that aprepitant is a broad-spectrum antitumor drug (against carcinomas, osteosarcoma, glioma, neuroblastoma, retinoblastoma, etc.) [48]. In vitro, aprepitant/fosaprepitant decreased tumor cell viability and increased cell death, whereas in vivo, it decreased both tumor volume and cell proliferation [82,83]. Aprepitant downregulates the expression of lymphoid enhancer-binding factor 1, c-myc and cyclin D1; arrests the G2 cell cycle leading to apoptosis in cancer cells; impairs the interaction of beta-catenin with Forkhead box M1 promoting the inhibition of the Wnt canonical pathway (which decreases super TOP/FOP and increases membrane stabilization of beta-catenin), and exerts an antitumor action (Figure 1) [20,84]. Moreover, aprepitant activates the caspase-3 dependent apoptotic pathway by altering the expression of the apoptosis-related genes, increases the sensitization of certain tumor cells to the cytotoxic action of vincristine/arsenic trioxide (ATO) and decreases the expression of liver stemness markers blood-brain barrier permeability and brain water content [46,85,86].

In comparison with solid tumors, fewer studies on the SP/NK-1R system has been reported in non-solid tumors [1,4,21,22,86,87]. In this section, the data on the involvement of that system in hematological malignancies, in particular in AML, and the possible treatment with NK-1R antagonists (aprepitant) are updated.

One study focused on the effect of aprepitant administered alone or in combination with a chemotherapeutic drug (ATO) against acute promyelocytic leukemia (APL)-derived NB4 cells and pre-B acute lymphocytic leukemia (ALL)-derived Nalm-6 cells [87]. In a dose-dependent manner, aprepitant showed against both cell lines a cytotoxic and antiproliferative action; the drug decreased both survival and the proliferative potential of these cells (Table 2) [87]. In comparison with NB4 cells, both DNA synthesis and viability were blocked in Nalm-6 cells by using higher concentrations of aprepitant (NB4: IC_50_, 7 μM; Nalm-6: IC_50_, 20 μM) [87]. Thus, in NB4 and Nalm-6 cells, aprepitant showed a differential sensitivity pattern. It was reported that this NK-1R antagonist, by increasing (p73/p21 expression) and blocking (c-myc expression), altered cell cycle and DNA replication rate, inhibiting the proliferation of NB4 cells (Table 2) [86]. Aprepitant, in combination with ATO, was more effective in inducing cytotoxic effects and the blockade of cell growth (NB4 cells), when compared to the administration of the NK-1R antagonist or ATO alone: thus, a synergistic anti-proliferative action was observed [87]. Coadministration of aprepitant and ATO induced an accumulation of NB4 cells in the sub-G1 phase and a decrease in DNA synthesis and in the proliferative capacity of these cells (by decreasing cell cycle S phase) (Table 2) [87]. It was demonstrated that the stimulatory effect of aprepitant on ATO-induced cytotoxicity was caused via caspase-3-dependent apoptosis: in NB4 cells, the number of apoptotic cells was higher when the coadministration of aprepitant-ATO was performed compared to the treatment of aprepitant alone (Table 2); in addition, aprepitant or ATO increased caspase-3 activity minimally but such activity increased markedly when NB4 cells were treated with aprepitant and ATO [87]. In other words, aprepitant activates the caspase-3-dependent apoptotic pathway and sensitizes NB4 cells to the cytotoxic effects mediated by vincristine or ATO (Table 2) [86]. In hematologic malignant cells, it has been reported that aprepitant via p-53-independent or dependent pathways increased a caspase-3-dependent apoptotic cell death (Figure 1) [63]. The blockade of the NF-κB pathway sensitized Nalm-6 cells to a lower concentration of aprepitant, and therefore, it seems that the effect of aprepitant is counteracted by an overactivation of the NF-κB pathway. Treatment with ATO showed a low inhibitory IκB degradation, but when combined with aprepitant, there was a decreased expression of phosphorylated-IκB and anti-apoptotic target genes of NF-κB, meaning that NK-1R antagonists increased the cytotoxic effect mediated by ATO by modulating the activity of NF-κB [87]. Thus, aprepitant, through the suppression of anti-apoptotic target genes of NF-κB, sensitized NB4 cells to ATO (Table 2) (Figure 1) [87]. Moreover, aprepitant and ATO increased the level of mRNA expression of pro-apoptotic targets (Bax, Bid, Bad, p21, p73) (Table 2) [87]. Thus, the data suggest that the ATO cytotoxic stimulatory effect, mediated by aprepitant, is due to an up-regulation of p73 transcription factor which can engage p21 and the NF-κB pathway through c-myc transcriptional suppression (Table 2) [87]. This has been suggested because it is known that the blockade of p73 activates NF-κB promoting cancer cell survival since apoptotic mechanisms were reduced [88,89]. In NB4 cells, ATO and aprepitant decreased the expression of c-myc, and it seems that the latter is the intermediate molecule through which p73 regulates NF-κB activity [90,91]. In this study, one of the most important findings was to show the molecular mechanisms involved in the resistance to aprepitant: the antitumor effect of this NK-1R antagonists is diminished when the NF-κB pathway is overactivated (Table 2) [87]. Another important finding is that aprepitant, in combination with chemotherapeutic drugs, was more effective in blocking cell growth and inducing cytotoxic effects and that this NK-1R antagonist exerted an antitumor action against acute promyelocytic leukemia (APL)-derived NB4 cells and pre-B acute lymphocytic leukemia (ALL)-derived Nalm-6 cells [87].

A recent study has shown that human AML cells (KG-1 (bone marrow) and HL-60 (peripheral blood acute promyelocytic leukemia)) express the NK-1R, and that this receptor mediates the anti-leukemic action of NK-1R antagonists (Table 1) [21]. The study also demonstrated that SP induced the proliferation of both AML cells in vitro (at nanomolar range) (Table 1), that four NK-1R antagonists (aprepitant (morpholine), L-733,060 (piperidine), L-732,138 (L-tryptophan), CP 96,345 (quinuclidine)) exerted, via the NK-1R, an anti-AML effect in a concentration-dependent manner (Table 2), but this growth inhibition observed in AML cells was marginal in lymphocytes, and that in a xenograft mouse model (with HL60 cells), fosaprepitant (administered intraperitoneally) increased the median survival from four (control group) to seven days (treated group) (Table 2) [21]. It is important to note that four NK-1R antagonists showing different chemical compositions performed an anti-AML effect, meaning that the antitumor action is not related to the chemical structures of the compounds but to their NK-1R affinities. Another important finding is that these AML cells overexpressed the truncated NK-1R when compared to healthy lymphocytes, and that NK-1R antagonists induced the death of AML cells by apoptosis (Table 2) [21]. In this study [21], four NK-1R antagonists were tested, but the highest antiproliferative action was observed when AML cell lines were treated with aprepitant or L-733,060. However, the proliferation of lymphocytes was not affected (even administering the highest dose of aprepitant) in the absence or presence of SP (Table 2). The IC_50_ for lymphocytes was ten-fold higher than that for AML; that is, lymphocytes were more resistant to NK-1R antagonists than AML cells and the damage induced by aprepitant in cancer cells was higher than that exerted in non-cancer cells (Table 2) [21]. This finding was also observed in fibroblasts and monocytes [92,93]; the molecular mechanisms involved in that resistance are currently unknown. A possible explanation could be the total number of NK-1Rs expressed by the cell and the number of full-length and truncated isoforms expressed. Nonetheless, the data show the safety of NK-1R antagonists and the specificity of the antitumor strategy applied, meaning that using these antagonists could avoid non desirable side-effects in cancer patients—non-desirable effects that are unfortunately observed using chemotherapeutic agents. Regarding the expression of the NK-1R, the study showed that two isoforms of the NK-1R (full-length and truncated) were present in AML cells and, in particular, the truncated form was two-fold higher than the full-length isoform (Table 1) [21]. In cancer cells, the truncated isoform is up-regulated and the full-length form is downregulated, and it has been suggested that a low level of the full-length isoform is responsible for the malignant phenotype of cancer cells [33]. However, in healthy cells, the truncated isoform was not expressed, whereas the full-length isoform was higher in AML cells (20-fold in KG-1 and 12-fold in HL-60) when compared to healthy cells (Table 1) [21]. Compared to lymphocytes, an increase in the expression of full-length *TAC1R* in both AML cells was not observed, but the latter cells showed a marked overexpression of the truncated splice variant (Table 1) [21]. It has been reported that the truncated isoform mediates tumor cell malignancy, increases the growth of these cells and promotes the synthesis of cytokines which exert growth-promoting effects (activates NF-κB that up-regulates the truncated isoform but slightly increases the full-length form), whereas the full-length isoform induces slow cancer cell growth [94,95,96]. By immunohistochemistry, a high level of NF-κB and cytokines has been reported in tumor tissues [36] and it has been shown that an increase in the expression of the NK-1R was observed after the activation of NF-κB by SP; thus, SP increases the expression of its natural ligand, the NK-1R [97].

In another recent study, a strong antitumor effect of aprepitant and L-733,060 has been demonstrated in vitro against acute myeloid leukemia cells (HL-60) (Table 2) [4]. Both NK-1R antagonists, in a concentration-dependent manner, induced apoptotic mechanisms in these cells and decreased the formation of colonies [4]. It has also been reported that necrotic mechanisms were observed in HL-60 cells, upgrading the concentration of the NK-1R antagonist (Table 2) [4]. In a concentration-dependent manner, both NK-1R antagonists blocked the formation of colonies (size, circularity, perimeter) of HL-60 myeloid leukemia cells (Table 2) [4].

In AML patients and human myeloid leukemia cell lines (NB4a, KG-1α, HL60, K562), a high expression of the NK-1R has been found in white blood cells and after blocking this receptor with NK-1R antagonists (aprepitant, SR-140,333), apoptotic mechanisms (in a concentration-dependent manner) were triggered by increasing mitochondrial reactive oxygen species (ROS) (Table 2) [1]. After analyzing the NK-1R gene expression, it was observed that AML patients showed an up-regulation of the NK-1R mRNA expression when compared to healthy controls (Table 1) [1]. Mitochondrial oxidative stress was due to a rapid Ca^++^ flux from the endoplasmic reticulum into mitochondria, leading to an impairment of the mitochondrial function and energy production (Table 2). Moreover, oxidative stress mechanisms can promote the activation of DNA damage pathways (e.g., ATR-CHK1 and ATM-CHK2 checkpoints) [97]. The blockade of the NK-1R with SR-140,333 increased the phosphorylation of CHK2 and ATM; this agrees with the activation of DNA damage pathways by oxidative stress and means that the cytotoxic effect of NK-1R antagonists is mediated by the production of ROS [1]. The administration of a ROS scavenger increased the viability of human myeloid leukemia cell lines in presence of NK-1R antagonists [1]. In contrast, SP did not promote the production of ROS in these cell lines [1]. SP, via the NK-1R, promotes weak mitochondrial and intracellular Ca^++^ fluxes, but NK-1R antagonists (aprepitant, SR-140,333) favor a rapid cytosolic/mitochondria Ca^++^ elevation in human myeloid leukemia cell lines (Table 1 and Table 2) [1]. The Ca^++^ transferred from the endoplasmic reticulum into the mitochondria acts as an apoptotic stimulus [98]. Thus, it seems that the blockade of the NK-1R by NK-1R antagonists (independent of the chemical structure) induce an endoplasmic reticulum/mitochondrial Ca^++^ overload, promoting a mitochondrial dysfunction that leads to the production of ROS and cell apoptosis. As the bone marrow of leukemia patients expands (due to an abnormal increase in the number of white blood cells) and leads to bone pain [99,100], the authors of the study reported another important finding: the blockade of the NK-1R exerted an antinociceptive effect in myeloid leukemia-induced bone pain by promoting apoptosis in leukemia cells and by counteracting inflammatory mechanisms in the bone/tumor microenvironment (Table 2) [1]. Pro-inflammatory mediators are increased in bone pain induced by cancer [101]. Bone marrow cells from mice transplanted with K562 cells showed a higher expression of cytokines (tumor necrosis factor-alpha, IL-1, IL-6), but treatment with SR-140,333 returned the expression to normal levels [1]. Moreover, this antagonist inhibited the expression of the three mentioned cytokines in K562 cells, and in mice inoculated with these cells and treated with SR-140,333, an up-regulation of pro-apoptotic proteins (cleaved caspase-3 and 8, Bim, Bax) was found [1]. In all of the AML patients studied, the expression of the NK-1R was observed, showing a weak (35%), moderate (41%) or strong (24%) immunoreactivity (located in the cytoplasm and cell membrane), whereas in healthy subjects this expression (always weak) was only observed in 8% of the population (Table 1) [1]. SP and NK-1R immunoreactivity was also studied in AML (HL-60, KG-1α, NB4) and chronic (K562) myeloid leukemia cells: all myeloid leukemia cell lines showed a moderate-strong immunoreactivity for both SP and the NK-1R, but in healthy subjects no immunoreactivity was found for both markers (Table 1) [1]. The presence of SP in myeloid leukemia cell lines suggests that the undecapeptide could be released from these cells and SP could consequently promote the release of cytokines into the microenvironment of the tumor and the proliferation of tumor cells (by an autocrine/paracrine mechanism), could regulate endothelial cells (expressing the NK-1R) favoring angiogenesis, plasma extravasation and granulocyte infiltration and could activate immune cells expressing the NK-1R (e.g., neutrophils, mast cells); that is, these mechanisms, mediated by SP, amplify the inflammatory response [1,9,102]. It is known that SP, through the NK-1R, is a crucial mediator in inflammatory mechanisms (vasodilatation, vascular permeability increase, edema formation, leukocyte infiltration) and that the undecapeptide promotes the synthesis of inflammatory cytokines (IL-1, IL-6, IL-12, tumor necrosis factor-alpha) by monocytes; in addition to favor an inflammatory response, the level of cytokines has been associated with an increased tumor progression [103,104,105]. Contrastingly, NK-1R antagonists decrease the expression of inflammatory cytokines, and thus, the inflammatory response [106]. It has been also reported that the viability of leukemia cells was reduced after the depletion of the NK-1R and that treatment with NK-1R antagonists induced an increase of apoptotic markers (annexin-V/propidium iodide), pro-apoptotic proteins (Bim, Bam, PARP, cleaved caspase-3 and 9) and the percentage of cells with sub-G1 content in these cells, whereas a decrease in anti-apoptotic proteins (Bcl-xL, Bcl-2) was also observed (Table 2) [1]. After treatment with NK-1R antagonists, a cell cycle arrest was observed in leukemia cells: a cell increase in G_0_/G_1_ phase was reported, as well as a significant decrease of cells in S phase, whereas the expression levels of CDC25A, CDK4, cyclin B1 and cyclin D1 were decreased and those of CDK inhibitors p16 and p21 were increased (Table 2) [1]. In a K562 xenograft mouse model, the antitumor effect of SR-140,333 (10 mg/kg, in situ injection) was tested, and a significant reduction of the tumor volume was observed without any serious side-effects [1]. One important finding was that neither SR-140,333 nor aprepitant exerted a proliferation-inhibitory effect against human normal hematopoietic cells, and no hemolytic toxicity was reported in human red blood cells (Table 2) [1]. This finding shows the selective action and safety of NK-1R antagonists against human myeloid leukemia cell lines. It has been also reported that SR-140,333 decreased the phosphorylation of downstream molecules of the mTOR complex 1 pathway (however, this pathway is not a key mediator in the cytotoxic effect promoted by SR-140,333; the decrease appears to be consequential to the mitochondrial dysfunction), increased the phosphorylation of p65 (a member of the NF-κB pathway) and inhibited the expression of myc, however the phosphorylation of ERK was not modified (Figure 1) [1].

Anthracyclines (doxorubicin, daunorubicin, epirubicin, idarubicin) are effective drugs used to treat many types of cancer, including AML [107]. However, an important adverse event can occur: the appearance of cardiotoxicity. The SP/NK-1R system mediates doxorubicin cardiotoxicity and chemoresistance; in fact, in tumor cells, aprepitant increases doxorubicin induced reduction of cell viability, ROS synthesis and cell death by apoptosis, whereas in normal cardiomyocytes, the same drug exerts opposite effects [108]. This means that aprepitant could promote beneficial effects by decreasing cardiotoxicity and increasing AML sensitivity to doxorubicin. This must be confirmed in future studies.

In summary, many data (in vitro and in vivo preclinical and clinical studies) show that the SP/NK-1R system is involved in AML and that NK-1R antagonists (aprepitant) is a promising antitumor strategy against this hematological malignancy. AML patients showed an up-regulation of the NK-1R mRNA expression when compared to healthy controls. Human AML cells express the NK-1R and, via this receptor, SP and NK-1R antagonists respectively exert a proliferative action or an anti-leukemic effect (apoptotic mechanisms are triggered by promoting oxidative stress via mitochondrial Ca^++^ overload). AML cell lines show immunoreactivity for both SP and the NK-1R, but in healthy subjects, no immunoreactivity was found. In AML cells, the truncated isoform is more expressed than the full-length form. Aprepitant also inhibits the formation of AML cell colonies and this NK-1R antagonist, when combined with chemotherapeutic drugs, is more effective in inducing cytotoxic effects and AML cell growth blockade: aprepitant, via suppression of anti-apoptotic target genes of NF-κB, sensitizes AML cells to chemotherapeutic drugs. NK-1R antagonists also exert an antinociceptive effect in myeloid leukemia-induced bone pain. The molecular mechanisms involved in the resistance to aprepitant were also demonstrated: the antitumor effect of this NK-1R antagonist is diminished when the NF-κB pathway is overactivated. This is important, because the activation of NF-κB by SP also increases the expression of the NK-1R. The data also show the safety and the high specificity of the NK-1R antagonist aprepitant to treat AML: the damage induced by aprepitant in cancer cells is higher than that exerted in non-cancer cells (the IC_50_ for lymphocytes is ten-fold higher than that for AML cells). Finally, aprepitant does not exert a proliferation-inhibitory effect against human normal hematopoietic cells and no hemolytic toxicity was reported in human red blood cells.

## 6. Conclusions

To date, in vitro and in vivo experiments have shown that the SP/NK-1R system, in a similar way, is involved in cancer progression in both solid (e.g., pancreas, lung, breast, osteosarcoma, retinoblastoma, glioma, neuroblastoma) and non-solid (AML) tumors. SP mediates a common mechanism for the proliferation of tumor cells. In cancer cells, the key point is the overexpression of the NK-1R, because the binding of SP to the NK-1R promotes beneficial effects in cancer cells (mitogenesis, favors cell migration and invasion, increases NK-1R transcription, induces glycogen breakdown (Warburg effect), anti-apoptotic action) and increases angiogenesis in bone marrow. For these reasons, the NK-1R in cancer treatment can be considered an important target, even more so when considering that it is a potential common therapeutic target for any type of cancer treatment. As tumor cells, including AML cells, overexpress the NK-1R, the use of NK-1R antagonists could be an excellent common antitumor strategy against AML, since it is a heterogeneous disease that is difficult to treat due to the characteristics of the cytogenetic subtypes and the potentially-differing morphological and clinical properties from person to person. However, it seems that in all AML cases, the NK-1R will be overexpressed in tumor cells. NK-1R antagonists counteract all of the aforementioned beneficial effects mediated by SP; these antagonists are well-tolerated and safe drugs that show excellent selectivity and potency against tumor cells. Aprepitant is a broad-spectrum antitumor drug against solid and non-solid tumors (e.g., AML), because it induces the death of tumor cells by apoptosis in vitro and in vivo experiments. Drug reprofiling or therapeutic switching is the use of an approved drug (e.g., aprepitant, used in clinical practice for preventing chemotherapy-induced vomiting and nausea) for a new indication (e.g., antitumor agent), and this means that the safety, metabolism, pharmacokinetic and contraindications of aprepitant are well-known and have been studied in depth. However, in future clinical studies (aprepitant reprofiling), to reach an antitumor effect, the dose and administration days of aprepitant must be increased in comparison to the current standard administered in clinical practice (which stands at 125 mg for the first day; 80 mg for the second day; and 80 mg for the third day). As reported previously, it seems that, to obtain a therapeutic antitumor action, an adequate dose (>20 mg/kg/day) must be administered daily and for a long period of time according to the response to treatment [109]. Overall, the current findings suggest that aprepitant inhibits AML cell growth and is a potential anti-leukemic drug, hence the use of NK-1R antagonists due to the involvement of the SP/NK-1R system, which, in cancer, opens an antitumor promising research line that must be confirmed in clinical trials. However, in depth studies in humans are needed, and phase 1 dose escalation studies with high doses of aprepitant must be performed to assess tolerability, and drug–drug interactions must be clearly established. Finally, clinical trials (against solid and non-solid tumors as AML), administering aprepitant as a single drug or in combination with chemotherapeutic drugs, could be carried out.

## Figures and Tables

**Figure 1 jcm-09-01659-f001:**
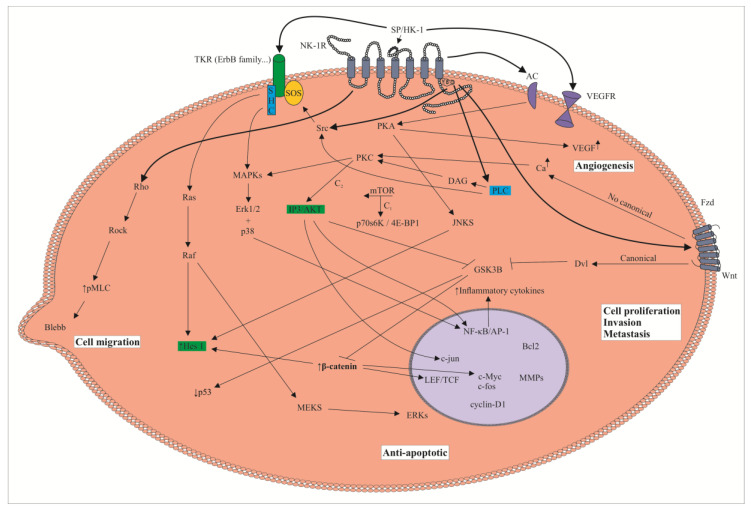
Tumor cell: signaling pathways downstream of the neurokinin-1 receptor (NK-1R). Substance P (SP), after binding to the NK-1R, promotes tumor cell proliferation and migration and an anti-apoptotic effect. In endothelial cells, SP via the NK-1R favors angiogenesis. NK-1R antagonists block these pathways and inhibit the effects mediated by SP on tumor and endothelial cells. 4E-BP 1: eukaryotic initiation factor 4E-binding protein 1; AC: adenylyl cyclase; AKT: protein kinase B; DAG: diacylglycerol; Dvl: dishevelled; ERKs: extracellular signal-regulated kinases; Fzd: Frizzled receptor; GSK3B: glycogen synthase kinase beta; HK-1: hemokinin-1; Hes 1: hairy and enhancer of split 1; IP3: inositol triphosphate; JNKS: c-Jun N terminal kinases; LEF/TCF: lymphoid enhancer-binding factor/transcription factor; MAPKs: mitogen-activated protein kinase; MEKS: mitogen-activated protein kinase kinases; MMPs: matrix metalloproteinases; mTOR: mammalian target of rapamycin; PKA: protein kinase A; PKC: protein kinase C; PLC: phospholipase C; pMLC: myosin light-chain kinase; p70s6K: p70 s6 kinase; TKR: tyrosine kinase receptor; VEGF: vascular endothelial growth factor; VEGFR: vascular endothelial growth factor receptor.

**Table 1 jcm-09-01659-t001:** The SP/NK-1R system in acute myeloid leukemia (AML) cells.

Leukemia cells express SP but not neurokinin A: related to leukemia and bone marrow fibrosis
SP increases the proliferation of bone marrow progenitors; neurokinin A exerts an opposite effect
AML cells express the NK-1R which mediates the antileukemic action of NK-1R antagonists
SP induces the proliferation of AML cells
Two NK-1R isoforms in AML cells: the truncated form is higher expressed than the full-length
AML cells: full-length expression is higher than in healthy cells, in which is not expressed
Compared with lymphocytes, AML cells overexpress the truncated splice variant
AML patients/cells: a high NK-1R expression is found in white blood cells
AML patients show an up-regulation of the NK-1R mRNA expression
SP, via the NK-1R, promotes a weak mitochondrial and intracellular Ca^++^ flux
All AML patients express the NK-1R. Healthy subjects: expression observed in 8% of population
AML cells express SP and NK-1R. Healthy subjects: no immunoreactivity

**Table 2 jcm-09-01659-t002:** Effects of aprepitant in AML cells.

In a dose-dependent manner exerts cytotoxic/antiproliferative effects and decreases AML cell survival/proliferative potential
Increases p73/p21 expression and alters cell cycle/DNA replication rate
Activates caspase-3-dependent apoptotic pathway
Aprepitant-ATO increase the number of AML apoptotic cells when compared to aprepitant alone
Aprepitant-ATO accumulate AML cells in sub-G1 phase and decrease DNA synthesis
Sensitizes AML cells to ATO via suppression of anti-apoptotic target genes of NF-κB
Aprepitant-ATO increase mRNA expression of pro-apoptotic targets (Bax, Bid, Bad, p21, p73)
Aprepitant-ATO decrease the expression of c-myc and regulates NF-κB activity
Diminished antitumor effect when the NF-κB pathway is overactivated
AML xenograft model: fosaprepitant increases the median survival 4–7 days
Induces AML cell death by apoptosis
A high dose of aprepitant does not affect lymphocytes proliferation
Exerts a higher damage in AML cells than in non-cancer cells
Blocks the formation of colonies of AML cells (size, circularity, perimeter)
Necrotic mechanisms are observed in AML cells upgrading aprepitant concentration
Increases mitochondrial reactive oxygen species
Mitochondrial oxidative stress is due to Ca^++^ flux from endoplasmic reticulum to mitochondria
Exerts an antinociceptive effect in myeloid leukemia-induced bone pain
Increases the level of apoptotic markers (annexin-V/propidium iodide)
Increases G_0_/G_1_ phase and decreases cells in S phase
No effect on human normal hematopoietic cells and no hemolytic toxicity on red blood cells

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
