# Peer review of "The Neurokinin-1 Receptor Antagonist Aprepitant, a New Drug for the Treatment of Hematological Malignancies: Focus on Acute Myeloid Leukemia"

_jcm, 2020, doi:10.3390/jcm9061659_

Round 1

Reviewer 1 Report

This is a paper looking at using neurokinin-1 receptor antagonists such as aprepitant as therapy for AML.  They present a fair amount of pre-clinical background as to the biologic rationale regarding exploring this area in the treatment of AML. It is overall a nice review of the topic and makes justification for further work in this area. 

Abstract:

1) Line 14:  I would change the wording that AML is “an incurable hematologic malignancy” as this is inaccurate.  Cure rates for AML are 50-60%, therefore it is curable in over half of people with it.  Agree that cure rates could be better and that novel therapeutics are needed.

2) Consider adding in the abstract that aprepitant is currently being used in many other cancers as an anti-nausea medication (which is what it is currently being used as routinely).

Main Body

1) Lines 42-46: again like the abstract I would remove the term incurable. Would suggest stating that it has very poor outcomes in adults with the majority not surviving 5 years (need to distinguish the difference in survival between children and older adults)

2) Lines 47-49:  agree that stem cell transplant is used for AML – but wording that it is used “chemotherapy has failed” is not correct.  We often use stem cell transplant upfront in treatment of AML (in first remission), especially where the patient has high risk features.

3) I would consider removing CML from this paper and focus only on AML.  CML is a different disease, although can accelerate into blast crisis and have an acute leukemia (AML or ALL) develop – at which time AML or ALL treatment is needed and where the outcomes are dismal.

4) line 143: “recent therapeutic strategies in which the SP/NK-1R is involved have shown excellent results”. Would change the word “excellent” with “promising”. The data you provide isn’t robust enough (one case report or aprepitant and a case series of Bi-DOTA-SP) to claim excellent results.

5) Line 152-153: stating “NK-R1 is an Achilles heel in cancer” is also an overstatement at this time. More evidence is needed.

6) Line 206: would change “most challenging hematologic malignancy” to a “very challenging hematologic malignancy” (as there are other malignancies with worse cure rates)

7) Line 233: Would suggest changing “aprepitant promotes a serious adverse event (febrile neutropenia)” to “One study has suggested an increase incidence of febrile neutropenia in pediatric osteosarcoma.”   In addition a meta-analysis (Br J Clin Pharmacol. 83(5):1108-1117, 2017 05) states that there was no increased fever neutropenia (that didn’t include this newer article), so I agree that further investigations are needed (esp if adding in to AML therapy that is already extremely myelosuppressive and with high risk of infection related deaths).

8) Reading this paper and the authors hypothesis that aprepitant has strong anti-tumor effects – is there any literature (in humans) that has shown a benefit of the use of aprepitant in outcomes when used as an anti-nausea agent? (ie did cure rates go up in solid tumors when the same protocols were used, but in the more recent era when aprepitant has been added in as an anti-nausea agent? (if any such literature exists it should be included).

9) Line 470: I would like previous comment for abstract be wary of calling aprepitant an Achilles heel against all cancer – especially given many patients with cancer already take aprepitant for nausea – why are so many of these patients not being cured if this is a true Achilles heel of cancer?    

10) Line 488: The dose needed for antineoplastic effects is incredibly high (instead of 125 mg per dose, it needs doses orders of magnitude higher (20 mg/kg). Are these doses safe and tolerable in humans (the product monograph for aprepitant for instance lists 600mg as the highest tolerable dose in healthy volunteers)

11) Line 492:  In follow up to my point above, before efficacy studies can be done in AML and solid tumors-  there needs to be Phase 1 dose escalation studies with high dose aprepitant to assess tolerability of high dose aprepitant.

12) Overall this is an interesting article that outlines the basic science as to why this pathway and medications such as aprepitant should be looked at from an anti-neoplastic point of view.  The use of aprepitant is very appealing based on pre-clinical work given that it can possibly help kill AML cells, while potentially reducing cardiotoxicity. It gives good scientific background on why further work in this area would be worth pursuing.  However, I think there needs to be a bit written on limitations of the fact that most of this is pre-clinical work only – that human studies are needed;  that investigations into whether high doses required in humans will be tolerated;  that drug-drug interactions will need to be investigated in not given as monotherapy.

Author Response

Comments and Suggestions for Authors

This is a paper looking at using neurokinin-1 receptor antagonists such as aprepitant as therapy for AML. They present a fair amount of pre-clinical background as to the biologic rationale regarding exploring this area in the treatment of AML. It is overall a nice review of the topic and makes justification for further work in this area.

Abstract:

1) Line 14: I would change the wording that AML is “an incurable hematologic malignancy” as this is inaccurate. Cure rates for AML are 50-60%, therefore it is curable in over half of people with it. Agree that cure rates could be better and that novel therapeutics are needed. The sentence has been corrected. See line 14.

2) Consider adding in the abstract that aprepitant is currently being used in many other cancers as an anti-nausea medication (which is what it is currently being used as routinely). This has been added. See lines 32 and 33.

Main Body

1) Lines 42-46: again like the abstract I would remove the term incurable. Would suggest stating that it has very poor outcomes in adults with the majority not surviving 5 years (need to distinguish the difference in survival between children and older adults). The sentence including “incurable” has been removed. According to the referee a new sentence has been added. See lines 43 and 44.

2) Lines 47-49: agree that stem cell transplant is used for AML – but wording that it is used “chemotherapy has failed” is not correct. We often use stem cell transplant upfront in treatment of AML (in first remission), especially where the patient has high risk features. The phrase “chemotherapy has failed” has been deleted. See lines 48 and 49.

3) I would consider removing CML from this paper and focus only on AML. CML is a different disease, although can accelerate into blast crisis and have an acute leukemia (AML or ALL) develop – at which time AML or ALL treatment is needed and where the outcomes are dismal. Sentences including CML have been removed from the new version. Thus, old references 7 and 8 have been removed and hence references have been re-numbered. See from lines 56 (text) and 532 (list of references; reference 7).

4) line 143: “recent therapeutic strategies in which the SP/NK-1R is involved have shown excellent results”. Would change the word “excellent” with “promising”. The data you provide isn’t robust enough (one case report or aprepitant and a case series of Bi-DOTA-SP) to claim excellent results. This has been corrected. See line 146.

5) Line 152-153: stating “NK-R1 is an Achilles heel in cancer” is also an overstatement at this time. More evidence is needed. The term has been deleted. See line 155.

6) Line 206: would change “most challenging hematologic malignancy” to a “very challenging hematologic malignancy” (as there are other malignancies with worse cure rates). This has been corrected. See lines 229 and 230.

7) Line 233: Would suggest changing “aprepitant promotes a serious adverse event (febrile neutropenia)” to “One study has suggested an increase incidence of febrile neutropenia in pediatric osteosarcoma.”   In addition a meta-analysis (Br J Clin Pharmacol. 83(5):1108-1117, 2017 05) states that there was no increased fever neutropenia (that didn’t include this newer article), so I agree that further investigations are needed (esp if adding in to AML therapy that is already extremely myelosuppressive and with high risk of infection related deaths). This has been corrected. See line 255.

8) Reading this paper and the authors hypothesis that aprepitant has strong anti-tumor effects – is there any literature (in humans) that has shown a benefit of the use of aprepitant in outcomes when used as an anti-nausea agent? (ie did cure rates go up in solid tumors when the same protocols were used, but in the more recent era when aprepitant has been added in as an anti-nausea agent? (if any such literature exists it should be included). To our knowledge, no study showing this beneficial effect has been reported. This probably occurs because the dose used as anti-emetic is very low to exert an anti-tumor effect. In cancer cells, the NK-1R is overexpressed, a higher dose of the antagonist is required. See lines 494-500 and reference 109.

9) Line 470: I would like previous comment for abstract be wary of calling aprepitant an Achilles heel against all cancer – especially given many patients with cancer already take aprepitant for nausea – why are so many of these patients not being cured if this is a true Achilles heel of cancer? The term has been changed by “an important target”. See line 480. See point 8: in our opinion, it is a question of dose.

10) Line 488: The dose needed for antineoplastic effects is incredibly high (instead of 125 mg per dose, it needs doses orders of magnitude higher (20 mg/kg). Are these doses safe and tolerable in humans (the product monograph for aprepitant for instance lists 600mg as the highest tolerable dose in healthy volunteers). In a case report, published by our group, the patient suffering from lung cancer was treated with aprepitant (1,140 mg/day for 45 days): the tumor mass disappeared and no serious side-effects were observed [50]. See lines 146-148 and reference 50 (lines 655-657). Obviously, more cases must be studied, but to our knowledge it is the first case published in which a high dose of aprepitant (as antitumor drug in combination with radiotherapy) was administered without side-effects. For this reason, we suggest the dose that must be administered to exert an antitumor action. In cancer cells, the overexpression of the receptor occurs and hence the dose administered must be high.

11) Line 492: In follow up to my point above, before efficacy studies can be done in AML and solid tumors- there needs to be Phase 1 dose escalation studies with high dose aprepitant to assess tolerability of high dose aprepitant. A new sentence has been added. See lines 503-507. See also next point.

12) Overall this is an interesting article that outlines the basic science as to why this pathway and medications such as aprepitant should be looked at from an anti-neoplastic point of view. The use of aprepitant is very appealing based on pre-clinical work given that it can possibly help kill AML cells, while potentially reducing cardiotoxicity. It gives good scientific background on why further work in this area would be worth pursuing. However, I think there needs to be a bit written on limitations of the fact that most of this is pre-clinical work only – that human studies are needed; that investigations into whether high doses required in humans will be tolerated; that drug-drug interactions will need to be investigated in not given as monotherapy. These ideas have been added. See lines 503-507.

Reviewer 2 Report

In sections 2, 3 and 4 the authors reported information about the SP/NK-1R system in tumor cells. However, they are not clear about the tumor type these observation were done, neither if these data derived from in vitro or in vivo studies. Although they would make generalizations about the utility to target the SP/NK-1R system in cancer, this is a limitation for the potential readers, due to the high tumor heterogeneity that might reduce the possibility to apply the same treatment strategy to different tumors (and this is particularly relevant between solid and hematological malignancies).

The manuscript is difficult to follow, especially in some parts, because it is full of information. I suggest to reduce details in brackets and to include at least one figure to summarize the signaling pathways and cellular processes in which the SP/NK-1R system is involved in tumor cells.

Both table 1 and 2 are not helpful: they should be more schematic.

Because the authors reported data about SP/NK-1R inhibition/expression in CML, AML and ALL, (section 5) in the introduction I suggest to remove the description of AML and CML, but to briefly introduce leukemia in general. For the same reason, I suggest to change the section 5 heading.

Because this manuscript reported both information about SP/NK-1R system in solid cancers and then in several leukemia subtypes (CML, AML and ALL), I suggest to soften the title by eliminating “Focus on Acute Myeloid Leukemia”.

Author Response

Comments and Suggestions for Authors

  1. In sections 2, 3 and 4 the authors reported information about the SP/NK-1R system in tumor cells. However, they are not clear about the tumor type these observation were done, neither if these data derived from in vitro or in vivo studies. Although they would make generalizations about the utility to target the SP/NK-1R system in cancer, this is a limitation for the potential readers, due to the high tumor heterogeneity that might reduce the possibility to apply the same treatment strategy to different tumors (and this is particularly relevant between solid and hematological malignancies). Since 2004, our group is investigating the antitumor action of non-peptide NK-1R antagonists on many human cancer cell lines (e.g., lung, pancreas, thyroid, retinoblastoma, breast, melanoma, osteosarcoma, neuroblastoma, glioma, carcinomas, non-solid tumors). See references 7, 8, 12, 18, 22, 27, 37, 40-43, 50, 60, 77, 83 and 109. Findings were obtained from in vitro and in vivo experiments as well as from a case report (reference 109). In these experiments several non-peptide NK-1R antagonists were also tested (aprepitant, fosaprepitant, L-733,060, L-732,138). In all cases, that is, independent of the human cancer type SP induced the proliferation of tumor cells, cancer cells overexpressed the NK-1R and all the NK-1R antagonists studied promoted, in a concentration-dependent manner, the death of tumor cells by apoptosis. In both solid and non-solid tumors, we observed the same previous findings. For this reason, we have included in the Ms sections 2 (predictive factor and biomarker), 3 (signaling pathways, overexpression of the NK-1R) and 4 (NK-1R is essential for the viability of tumor cells). Thus in all three sections, we up-date the current data on the SP/NK-1R system in cancer Why? Because we want to remark that the mechanisms mentioned are common to all cancer cells and hence using NK-1R antagonists the same antitumor strategy can be developed for any cancer type. In these sections, we want also to remark the crucial role that plays the SP/NK-1R in cancer (including both solid and non-solid tumors). In summary, according to our experience, in sections 2-4 the data are shown in a general way because all these main mechanisms are common for any cancer type. Later, in section 5 the data reported (see below point 4) are exclusively from AML. A summary of the above ideas appears in lines 169-172 and 196-200. In addition, according to the suggestion of the referee, in lines 104, 113, 141, 146, 148, 166 and 190 results from patients and from in vitro and in vivo experiments have been mentioned.

  1. The manuscript is difficult to follow, especially in some parts, because it is full of information. I suggest to reduce details in brackets and to include at least one figure to summarize the signaling pathways and cellular processes in which the SP/NK-1R system is involved in tumor cells. Figure 1 has been added. See lines 71, 81, 98, 106, 125, 130, 135, 138, 145, 156, 158, 161, 172, 173-186, 199, 208, 226, 242, 268, 299, 306 and 442. The technical assistance of Dr. M Rosso and Mr Javier Muñoz has been mentioned. See lines 514-515.

  1. Both table 1 and 2 are not helpful: they should be more schematic. This has been done. Sentences have been shortened. Now the results are more clearly expressed. See lines 245 (Table 1) and 322 (Table 2).

  1. Because the authors reported data about SP/NK-1R inhibition/expression in CML, AML and ALL, (section 5) in the introduction I suggest to remove the description of AML and CML, but to briefly introduce leukemia in general. For the same reason, I suggest to change the section 5 heading. According to the suggestion of the referee, those sentences referring to ALL and CML in the old Ms (in Introduction and Section 5) have been removed in the new ones, as well as old reference 109. Thus, all the information appearing in Section 5 is exclusively from AML. See line 228. For this reason, the section 5 heading has remained in the new version. References have been re-numbered.

  1. Because this manuscript reported both information about SP/NK-1R system in solid cancers and then in several leukemia subtypes (CML, AML and ALL), I suggest to soften the title by eliminating “Focus on Acute Myeloid Leukemia”. Due to the reasons mentioned in points 1 and 4, we think that “Focus on Acute Myeloid Leukemia” must appear in the title. See line 4.

     We hope that these amendments are sufficient to make the paper acceptable for publication.

Sincerely yours,

Miguel Muñoz and Rafael Coveñas